# Estimating Glomerular Filtration Rate from Serum Myo-Inositol, Valine, Creatinine and Cystatin C

**DOI:** 10.3390/diagnostics11122291

**Published:** 2021-12-07

**Authors:** Frank Stämmler, Marcello Grassi, Jeffrey W. Meeusen, John C. Lieske, Surendra Dasari, Laurence Dubourg, Sandrine Lemoine, Jochen Ehrich, Eric Schiffer

**Affiliations:** 1Department of Research and Development, Numares AG, 93053 Regensburg, Germany; frank.staemmler@numares.com (F.S.); marcello.grassi@numares.com (M.G.); 2Department of Laboratory Medicine and Pathology, Mayo Clinic, Rochester, MN 55905, USA; meeusen.jeffrey@mayo.edu (J.W.M.); lieske.john@mayo.edu (J.C.L.); 3Division of Nephrology and Hypertension, Mayo Clinic, Rochester, MN 55905, USA; 4Division of Endocrinology, Diabetes, Metabolism, and Nutrition, Mayo Clinic, Rochester, MN 55905, USA; dasari.surendra@mayo.edu; 5Service d’Explorations Fonctionnelles Rénales et Métaboliques, Hôpital Edouard Herriot, 69437 Lyon, France; laurence.dubourg@chu-lyon.fr (L.D.); sandrine.lemoine@chu-lyon.fr (S.L.); 6Children’s Hospital, Hannover Medical School, 30625 Hannover, Germany; ehrich.jochen@mh-hannover.de

**Keywords:** glomerular filtration rate, eGFR, filtration markers, metabolite, NMR, chronic kidney disease, CKD, eGFR equation, serum creatinine, cystatin C

## Abstract

Assessment of renal function relies on the estimation of the glomerular filtration rate (eGFR). Existing eGFR equations, usually based on serum levels of creatinine and/or cystatin C, are not uniformly accurate across patient populations. In the present study, we expanded a recent proof-of-concept approach to optimize an eGFR equation targeting the adult population with and without chronic kidney disease (CKD), based on a nuclear magnetic resonance spectroscopy (NMR) derived ‘metabolite constellation’ (GFR_NMR_). A total of 1855 serum samples were partitioned into development, internal validation and external validation datasets. The new GFR_NMR_ equation used serum myo-inositol, valine, creatinine and cystatin C plus age and sex. GFR_NMR_ had a lower bias to tracer measured GFR (mGFR) than existing eGFR equations, with a median bias (95% confidence interval [CI]) of 0.0 (−1.0; 1.0) mL/min/1.73 m^2^ for GFR_NMR_ vs. −6.0 (−7.0; −5.0) mL/min/1.73 m^2^ for the Chronic Kidney Disease Epidemiology Collaboration equation that combines creatinine and cystatin C (CKD-EPI_2012_) (*p* < 0.0001). Accuracy (95% CI) within 15% of mGFR (1-P15) was 38.8% (34.3; 42.5) for GFR_NMR_ vs. 47.3% (43.2; 51.5) for CKD-EPI_2012_ (*p* < 0.010). Thus, GFR_NMR_ holds promise as an alternative way to assess eGFR with superior accuracy in adult patients with and without CKD.

## 1. Introduction

Renal function is generally assessed by estimating the glomerular filtration rate (GFR) using endogenous filtration markers, most often serum creatinine. Equations to estimate GFR (eGFR) are widely used in routine medical care, and generally include age, sex, and race together with creatinine and/or cystatin C [1,2,3,4,5,6]. The current Kidney Disease Improving Global Outcomes (KDIGO) guideline recommends a first-line determination of eGFR based on serum creatinine, and a confirmatory testing based on equations using cystatin C (either alone or in combination with creatinine) [7,8]. Yet, these equations present limitations, either under- or overestimating tracer measured GFR (mGFR) in various patient groups [4,5,6], such as liver cirrhosis, transplant recipients, patients with extreme body composition, or when accuracy of GFR estimation is of special importance, such as potential kidney donors or pharmacokinetic drug dosing in patients with impaired renal function [7,8]. Generally, filtration markers such as creatinine are influenced by non-GFR determinants. It has been recently acknowledged that the use of multiple filtration markers, such as metabolites, in combination with serum creatinine might be one way to overcome these limitations [4,6,9,10,11].

With this in mind, we recently employed a nuclear magnetic resonance spectroscopy (NMR) based metabolomics approach to identify a ‘metabolite constellation’ (GFR_NMR_) to reflect more accurately GFR among individuals with and without chronic kidney disease (CKD). This proof-of-concept study, integrating myo-inositol as marker for reduced filtration, valine as indicator of acid-base metabolism, and dimethyl sulfone as marker of oxidative stress in combination with serum creatinine, allowed accurate assessment of eGFR [12]. The observed accuracy exceeded that of serum creatinine and of serum cystatin C alone, and matched the accuracy of their combination in pediatric, adolescent, adult, and geriatric patients with various nephrological conditions [12].

In the present study, we further refined this approach with the goal to design a more accurate and robust eGFR equation, relative to tracer measured GFR as gold standard reference, for an adult population, which encompasses the highest risk group for CKD [13,14].

## 2. Materials and Methods

### 2.1. Study Design and Participants

Bio-banked serum samples from adult individuals ≥ 18 years old from Rochester (MN, USA), Lyon (France) [12], and Berlin (Germany) [15] were used for NMR metabolite quantification. All individuals gave informed consent before undergoing GFR measurement. This study was approved by the respective institutional review board at each institution in adherence to the Declaration of Helsinki (Mayo Clinic IRB# 19-003513, dated 16 May 2019). Samples were stored at −80 °C and underwent no more than one freeze-thaw cycle before NMR analysis. Qualified NMR spectra were obtained from 1855 serum samples in total. Samples underwent partitioning into “Development”, “Internal Validation” and “External Validation” datasets (Table 1) stratified by mGFR range, liver disease status, sex and clinical indication. The development set was used for equation formulation, training and pre-selection. The internal validation set was applied for selection and internal testing of pre-selected candidate equations. The external validation set was used for confirmation of performance on an independent dataset. The development and the external validation sets consisted of samples from Rochester (*n* = 816 and *n* = 600, respectively) with a homogenous reference standard to minimize potential reference bias, whereas the internal validation set was populated with samples from all three centers (Rochester, *n* = 269; Lyon and Berlin, *n* = 170) with heterogeneous reference methods to maximize generalization of the selected equation (Table 1). This partitioning was supported by the fact that we observed larger bias of the CKD-EPI equation in samples from Lyon and Berlin, compared to the cohort from Rochester. Hence, the applied partitioning appeared particularity suited to minimize the risk to overjudge superiority of models against CKD-EPI benchmarks due to a systematic bias in the external validation set by samples from Lyon and Berlin.

### 2.2. Laboratory Methods

#### 2.2.1. mGFR, Serum Creatinine and Cystatin C Measurements

The study samples contained iohexol [16], inulin [17], ^51^Cr-EDTA [18] or iothalamate [19] as a reference standard for GFR measurements (mGFR). All mGFR methods were reported to have sufficient accuracy compared with inulin clearance [20]. Measured GFR was normalized to body surface area according to the Dubois equation (body surface area = height^0.725^ × weight^0.425^ × 0.007184) and expressed as milliliter per minute per 1.73 m^2^ body surface (mL/min/1.73 m^2^). Enzymatic serum creatinine measurements were performed with methods traceable to the National Institute of Standards and Technology and were isotope-dilution mass spectrometry calibrated [21]. Serum cystatin C measurements were performed at Mayo Clinic using Gentian Cystatin C Immunoassay ERM-DA471/IFCC standardized (Gentian ASA, Moss, Norway). The Berlin cohort was measured at Labor Limbach Heidelberg using the PENIA N Latex^®^ assay on the BN™ II System (Siemens Health Care Diagnostics, ex-Dade-Behring, Marburg, Germany). For samples from Lyon cystatin C was measured with the Human Cystatin C ELISA from Biovendor (calibrated to standard reference material ERM-DA471/IFCC, Laborarztpraxis van de Loo, Schwäbisch Gmünd, Germany).

#### 2.2.2. eGFR

GFR_NMR_ was compared to the standard serum creatinine-based and/or cystatin C-based eGFR equations according to KDIGO recommendations [7,8,22]. Serum creatinine-based eGFR was calculated using the CKD-EPI 2009 creatinine equation (CKD-EPI_2009_) [1], the CKD-EPI 2012 cystatin C equation [2] was used for calculating eGFR from cystatin C (CKD-EPI_Cys_) and the 2012 CKD-EPI creatinine–cystatin C equation [2] was applied to calculate eGFR from both (CKD-EPI_2012_). For an age-independent serum creatinine-based eGFR, the European Kidney Function Consortium (EKFC) equation [3] was used.

#### 2.2.3. NMR Analysis and Biomarker Quantification

NMR analysis was performed as described elsewhere [12]. Briefly, 630 µL serum were mixed with 70 µL of Axinon^®^ serum additive solution and 600 µL were transferred to a 5 mm NMR tube. Samples were pre-heated at 37 °C for 7.5 min before NMR measurement in a Bruker Avance III 600 MHz and a 5 mm PATXI probe equipped with automatic Z gradients shimming. A modified version of the CPMG pulse sequence was used [23]. ^1^H-NMR spectra (Appendix A) were recorded using a spectral width of 20 ppm, with a recycling delay of 1.5 s, 16 scans and a fixed receiver gain of 50.4. A cycling time d2 of 8 ms was used together with a corresponding T2 filter of 112 ms. The mixing time τ between two consecutive spin echoes was 400 µs. NMR data were automatically phase- and baseline-corrected using the lactate doublet at 1.32 ppm as reference.

Metabolite quantification used curve-fitted pseudo-Voigt profiles, as previously described [12]. For analytical validation of markers (creatinine, creatine, dimethylamine, dimethyl sulfone, glycerol, isoleucine, leucine, myo-inositol, and valine), precision, linearity and bias were analyzed as described in detail elsewhere [12]. Total analytical imprecision for all markers was validated to be below 15%. Higher imprecision of <20% was only accepted for serum levels at limits of quantification of approximately 10 µmol/L. For all metabolites, Pearson correlation was verified to be >0.95. The NMR platform allowed accurate and precise quantification of the serum biomarkers over a linear range covering physiological and pathophysiological levels.

### 2.3. Biostatistical Methods

All calculations, model training, performance evaluation and statistical tests were performed within R 3.5.3 [24]. Most metrics were calculated with the ModelMetrics [25] and auditor [26] packages. Data structures were handled with data.table [27] and archivist [28] packages. Bootstrap procedures were implemented via the boot package [29,30]. Visualization was performed with ggplot2 [31] and auditor [26].

#### 2.3.1. Metrics for Performance Evaluation and Benchmarking

Bias was assessed by median signed bias and the significance of differences was calculated via the Wilcoxon-signed rank test [32,33]. Precision was evaluated by the interquartile range (IQR), whereas significance of differences was assessed via the bootstrap method. Accuracy of eGFR was assessed by the percentage of samples with an eGFR prediction within 30% (P30), 20% (P20) or 15% (P15) of mGFR. Accuracy data were expressed as 1-P30, 1-P20 and 1-P15, respectively, representing the percentage of samples outside the given tolerance range relative to mGFR. Accordingly, the lower the 1-P30, 1-P20 and 1-P15 values, the higher the eGFR accuracy. Statistical significance of differences in P30, P20 and P15 accuracy was assessed by the McNemar’s test [34]. Error was reported as mean absolute error (MAE) and confirmed by root-mean-square logarithmic error (RMSLE; not shown). Since MAE has the same unit as mGFR and eGFR, it is more easily interpretable than RMSLE. Furthermore, MAE is unambiguous compared to RMSLE [35]. Significance of differences in MAE was assessed via the Wilcoxon-signed rank test [32,33]. In all analyses, *p*-values ≤ 0.05 were considered statistically significant.

In order to evaluate improvement in CKD staging gained by using GFR_NMR_ vs. CKD-EPI_2009_ or CKD-EPI_2012_ relative to a gold standard (mGFR-based CKD staging), we calculated the net reclassification index (NRI) [36,37,38]. NRI was calculated as the difference between the percentages of correctly reclassified and incorrectly reclassified samples by GFR_NMR_ vs. CKD-EPI_2009_ or CKD-EPI_2012_.

#### 2.3.2. Development of Equations and Model Training

Model formulas were constructed from a feature pool consisting of 10 serum metabolites: nine measured by NMR (creatinine, creatine, dimethylamine, dimethyl sulfone, glycerol, isoleucine, leucine, myo-inositol, and valine), and one measured by clinical analyzer (cystatin C). From this pool of 10 parameters, each could contribute either untransformed or natural log-transformed, which increased the pool of possible features for a model building up to 20 parameters. To guide the construction of the model formulas, we introduced a set of four constraints. First, each model was allowed to include two to five features. Second, a maximum of one out of five features could be an interaction term. Third, if an interaction was part of a given model, at least one of the parameters involved had to be already included in the model as a main effect. Fourth, no parameter was allowed to occur with two different transformations of itself in the same model (e.g., no co-occurrence of creatinine and log-transformed creatinine within the same model). All possible model formulas under these constraints were subsequently trained by linear regression with mGFR or log-transformed mGFR as response on the development datasets. Furthermore, a five-time repeated fivefold cross-validation was performed for all models.

#### 2.3.3. Selection, Model Engineering and Internal Validation of Equations

The selection of candidate models went through four different selection stages: a pre-filtering phase, a model selection phase, a model engineering phase, and a final selection phase.

Instead of selecting the best performing models on a certain key performance indicator (KPI), a stepwise deselection of models not fulfilling a minimum of performance over a set of multiple KPIs was applied. The pre-filtering phase was performed on the training performance in the development dataset in 14 iterative steps. Filtered-out models included: models performing poorly with regard to MAE and P30, models showing unstable coefficient of variation for the MAE in cross-validation, models that were not able to predict the upper range of the GFR (i.e., GFR ≥ 90), or models with high heteroscedasticity in the residuals. With further model optimizations in mind, we elected to only keep models that were comparable enough to current benchmarks to benefit from the planned optimizations. Finally, we closely inspected related models. For nested models, for instance, we tested whether the additional parameter was really providing significant improvement (H_1_) or not (H_0_) (Vuong’s test) [39,40]. To further reduce redundancy, we removed the worst performers in terms of MAE and P20 within closely related models.

The second model selection phase was performed on the remaining models from the pre-filtering phase and aimed to remove low performing models in portions of the internal validation dataset. Hereto, the internal validation dataset was further split up into three subsets, one of which was not to be used in this phase. The selection phase consisted of eight iterative filtering steps, with the goal to focus on specific CKD etiologies and CKD stages. All decisions on deselection were guided by subject matter experts.

The remaining models after the second selection phase were then subjected to several modifications during the model engineering phase. These modifications were aimed at the contributions of creatinine and cystatin C in the model formulas. Similarly to CKD-EPI formulas [2,3], we found in initial experiments that adding a sex dependency and a cutoff dependency to creatinine and cystatin C (i.e., cutoffs for high and low serum concentrations) could be beneficial for more accurate prediction of eGFR. Allowing different coefficients based on sex as well as on creatinine and cystatin C levels was therefore tested. Moreover, age was added as a linear term in a separate step.

As the number of models resulting from the model engineering phase was increased again, a third and last selection phase was applied. The best metabolite constellation that optimized P20, P30 and MAE was selected on the full internal validation dataset, while considering the complexity of the models after model engineering.

#### 2.3.4. External Validation of the New Equation

Final model performance was tested on an independent dataset. All KPIs were assessed on the external validation dataset (*n* = 600; Table 1). For all KPIs, 95% confidence intervals (CI) were calculated via bootstrap (percentile method, *n* = 1000). All reported *p*-values were adjusted for multiple testing according to Benjamini-Hochberg [18,19].

## 3. Results

### 3.1. Characteristics of Participants

The clinical characteristics of the development, internal validation and external validation datasets are shown in Table 1. Measured GFR (mGFR) was comparable in the three datasets (mean [±SD] of 67 [±28], 67 [±8] and 68 [±27] mL per minute per 1.73 m^2^ of body-surface area, respectively). Mean ± SD [range] age in the development, internal validation and external validation datasets was 55 ± 14 (18–86), 58 ± 15 (18–88) and 56 ± 14 (19–88), respectively. Sex and CDK stage distributions were comparable in the three datasets (Table 1).

### 3.2. Formulation and Selection of Candidate Equations

Systematic, exhaustive modelling resulted in the formulation of 487,408 distinct equation candidates. In 14 iterative steps solely based on performance characteristics in the development dataset, pre-filtering reduced the number of candidate equations down to 44. By verification of generalizability of their observed performance characteristics into the internal validation set, the set of 44 candidate equations was further selected down to five candidate equations. Manual fine-tuning of these five candidate equations resulted in 49 engineered equation variants. Selection of the new GFR_NMR_ equation from those 49 engineered variants was based on the comparability of performance levels in the development and the internal validation datasets. The resulting GFR_NMR_ equation was based on myo-inositol, valine, creatinine and cystatin C serum levels plus age and sex. Table 2 shows the GFR_NMR_ equation according to cystatin C cutoff levels and sex.

### 3.3. The New GFR_NMR_ Equation and Its Performance in the External Validation Dataset

The performance of the GFR_NMR_ equation was evaluated in the external validation set and compared to that of other equations (CKD-EPI_2009_, CKD-EPI_2012_, CKD-EPI_Cys_ and EKFC; [1,2,3]) (Table 3). The GFR_NMR_ equation showed the best performance in the overall external validation set in terms of median bias (median difference, 0.0 mL/min/1.73 m^2^), precision (interquartile range [IQR] of the difference, 13 mL/min/1.73 m^2^), error (mean absolute error [MAE], 10 mL/min/1.73 m^2^) and accuracy (1-P15, 38.8%; 1-P20, 28.5%; 1-P30, 12.8%). The GFR_NMR_ equation also performed best in the eGFR range < 60 mL/min/1.73 m^2^, which is defined as ‘decreased GFR’ according to the KDIGO guideline [7,8] and is therefore clinically relevant (Table 3).

The overall precision of GFR_NMR_ (IQR 13 mL/min/1.73 m^2^) was significantly greater than that of CKD-EPI_2009_ (16.4 mL/min/1.73 m^2^; *p* = 0.0167), CKD-EPI_Cys_ (18.1 mL/min/1.73 m^2^; *p* = 0.0167) and EKFC (17.0 mL/min/1.73 m^2^; *p* = 0.0167), but not significantly greater than that of CKD-EPI_2012_ (14.0 mL/min/1.73 m^2^; *p* = 0.7333) (Table 3). The overall error for GFR_NMR_ (MAE 10 mL/min/1.73 m^2^) was significantly lower than that for CKD-EPI_2009_ (11.9 mL/min/1.73 m^2^; *p* < 0.0001), CKD-EPI_2012_ (11.1 mL/min/1.73 m^2^; *p* < 0.0001), CKD-EPI_Cys_ (13.3 mL/min/1.73 m^2^; *p* < 0.0001), and EKFC (12.8 mL/min/1.73 m^2^; *p* < 0.0001) (Table 3).

The overall median bias of GFR_NMR_ (0.0 mL/min/1.73 m^2^) was significantly lower than that of CKD-EPI_2009_ (−4.0 mL/min/1.73 m^2^; *p* < 0.0001), CKD-EPI_2012_ (−6.0 mL/min/1.73 m^2^; *p* < 0.0001), CKD-EPI_Cys_ (−7.0 mL/min/1.73 m^2^; *p* < 0.0001), and EKFC (−6.5 mL/min/1.73 m^2^; *p* < 0.0001) (Table 3). When evaluated according to the eGFR range, the median bias of GFR_NMR_ was closest to zero between 15–29, 30–59 and 60–89 mL/min/1.73 m^2^, among all eGFR equations (Figure 1A). Moreover, the 95% confidence interval (CI) of GFR_NMR_ was the narrowest in the 30–59 mL/min/1.73 m^2^ eGFR range. The median bias < 15 mL/min/1.73 m^2^ showed higher 95% CI for all equations, as expected in this range of lower measurement precision of mGFR, and probably due to the low *n* values (*n* = 3 to 6). At eGFR ≥ 90 mL/min/1.73 m^2^, CKD-EPI_2012_ showed the lowest median bias, closely followed by CKD-EPI_2009_ and GFR_NMR_ (Table 3). However, GFR_NMR_ showed the lowest 95% CI among all formulas in that upper eGFR range (Table 3 and Figure 1A). Overall, while the median bias of GFR_NMR_ was consistently close to zero across all eGFR ranges, that of other equations indicated a consistent underestimation of GFR. In addition, the CKD-EPI_Cys_ equation showed fluctuating results, underestimating GFR < 90 mL/min/1.73 m^2^ while overestimating it ≥ 90 mL/min/1.73 m^2^ (Figure 1A). Finally, the low bias achieved by GFR_NMR_ was emphasized by comparing the distribution of the absolute and relative bias values of the eGFR equations in the external validation set (Appendix A).

The overall P30 accuracy of GFR_NMR_ was significantly higher than that of CKD-EPI_Cys_ (1-P30, 12.8% vs. 22.8% respectively; *p* < 0.0001), but not significantly higher than that of the other equations (CKD-EPI_2009_, CKD-EPI_2012_ and EKFC; Table 3). On the other hand, the overall accuracy at a narrower margin of error (P20 and P15) was significantly higher for GFR_NMR_ than for all other formulas (1-P20: 28.5% for GFR_NMR_ vs. 37.2% for CKD-EPI_2009_ (*p* < 0.001), 33.3% for CKD-EPI_2012_ (*p* < 0.0001), 44.0% for CKD-EPI_Cys_ (*p* < 0.0001), 40.5% for EKFC (*p* < 0.0001); 1-P15: 38.8% for GFR_NMR_ vs. 49.5% for CKD-EPI_2009_ (*p* < 0.0001), 47.3% for CKD-EPI_2012_ (*p* < 0.010), 56.0% for CKD-EPI_Cys_ (*p* < 0.0001) and 53.0% for EKFC (*p* < 0.0001) (Table 3).

When evaluated according to the eGFR range, the P20 accuracy was best (lowest 1-P20 value) for GFR_NMR_ between 15 and 59 mL/min/1.73 m^2^ (Figure 1B), which represents the most clinically relevant GFR range when assessing CKD stage classification [7,8]. In that range, CKD-EPI_Cys_ exhibited the lowest accuracy (highest 1-P20 value). Notably, GFR_NMR_ clearly outperformed all other equations in the 30–59 mL/min/1.73 m^2^ eGFR range, with the lowest 1-P20 and the narrowest 95% CI (Figure 1B). At eGFR > 60 mL/min/1.73 m^2^, GFR_NMR_ and CKD-EPI_2012_ performed equally well (similarly low 1-P20 and narrow 95% CI). The 1-P20 values < 15 mL/min/1.73 m^2^ could not be interpreted due to the too low *n* values (*n* = 3 to 6).

Figure 2 highlights the improved accuracy of GFR_NMR_ over all other equations for error tolerance cutoffs < P30. While at P30, GFR_NMR_ performed comparably to CKD-EPI_2012_, GFR_NMR_ outperformed all other evaluated equations below this arbitrary cutoff (Figure 2).

### 3.4. Performance of the New GFR_NMR_ Equation in Subpopulations of the External Validation Dataset

A subgroup analysis according to age (<40, 40–65 and >65 years old) showed significantly reduced median bias and MAE for the new GFR_NMR_ equation compared to all other equations in age groups 40–65 and >65 years (Appendix A). P15 and P20 accuracy of GFR_NMR_ was also significantly improved compared to all other equations in the >65-year-old group (Appendix A). GFR_NMR_’s P15 and P20 accuracy in the 40–65 age’ group were comparable to those of CKD-EPI_2012_ but significantly higher than those achieved by other equations (Appendix A).

A subgroup analysis according to sex confirmed that the GFR_NMR_ equation outperformed other equations in terms of bias, MAE and P15 accuracy in both males and females (Appendix A). GFR_NMR_’s P20 accuracy was comparable to that of CKD-EPI_2012_ but significantly higher than that of other equations in males and females (Appendix A).

A subgroup analysis according to body mass index (BMI; groups: <20, 20–25, 25–30 and >30 kg/m^2^) revealed that the GFR_NMR_ equation had a significantly lower median bias than all other equations for the 20–25, 25–30 and >30 kg/m^2^ BMI subgroups (Appendix A). GFR_NMR_’s MAE was comparable to that of CKD-EPI_2012_ but significantly higher than that of other equations in the 20–25 and 25–30 kg/m^2^ BMI subgroups, while GFR_NMR_’s MAE outperformed all equations in the >30 kg/m^2^ BMI group (Appendix A). GFR_NMR_’s P15 accuracy was comparable to that of CKD-EPI_2012_ but significantly higher than that of other equations in the 20–25 and 25–30 kg/m^2^ BMI subgroups, while it was comparable to that of CKD-EPI_2009_ but significantly higher than that of other equations in the >30 kg/m^2^ BMI subgroup (Appendix A).

We also performed an exploratory analysis according to ethnicity. Ethnicity was determined by patient questionnaire and all patients who self-identified as African, African American or Black were grouped as black, and all others were grouped as non-black. This analysis suggested comparable bias distribution for GFR_NMR_ in black vs. non-black individuals (Appendix A).

Altogether, these subgroup analyses underlined the improved performance of GFR_NMR_ in clinically relevant populations, such as individuals >65 years of age or with elevated BMI (>25 kg/m^2^), compared to other equations.

### 3.5. Evaluation of the Potential Clinical Benefit of the New GFR_NMR_ Equation

Finally, we evaluated the potential clinical utility of the new GFR_NMR_ equation, by assessing the proportion of correct and incorrect CKD staging reclassification by GFR_NMR_ vs. the equations recommended by KDIGO, CKD-EPI_2009_ and CKD-EPI_2012_ [7,8] (Table 4). The NRI, corresponding to the difference between correctly and incorrectly reclassified CKD staging, calculated on the overall external validation dataset, was 8.0% (22.2% correctly and 14.2% incorrectly reclassified) for GFR_NMR_ vs. CKD-EPI_2009_ and 7.6% (17.8% correctly and 10.2% incorrectly reclassified) for GFR_NMR_ vs. CKD-EPI_2012_ (Table 4). This indicates that the GFR_NMR_ equation globally reclassified more correctly CKD staging than the respective CKD-EPI equations did (Table 4).

When evaluated according to the eGFR range, GFR_NMR_ more correctly reclassified CKD staging for eGFR ≥ 45 mL/min/1.73 m^2^, i.e., for ranges 45–59 (CKD stage G3a), 60–89 (CKD stage G2) and ≥90 (CKD stage G1) mL/min/1.73 m^2^, with NRI of 9.1, 14.9 and 3.7, respectively, vs. CKD-EPI_2009_, and NRI of 9.9, 14.5 and 6.7, respectively, vs. CKD-EPI_2012_. In the 30–44 mL/min/1.73 m^2^ eGFR range (CKD stage G3b), CKD-EPI_2009_ performed better than GFR_NMR_ (NRI −3.8), while GFR_NMR_ and CKD-EPI_2012_ performed equally well (NRI 0.0). In the 15–29 mL/min/1.73 m^2^ eGFR range (CKD stage G4), GFR_NMR_ performed better than CKD-EPI_2009_ in CDK staging reclassification (NRI 5.6), while CKD-EPI_2012_ performed better than GFR_NMR_ (NRI −16.6). As before, no proper evaluation could be conducted for eGFR ≤ 15 mL/min/1.73 m^2^ (CKD stage G5), due to too low sample number (*n* = 4) (Table 4).

## 4. Discussion

GFR_NMR_ as described here outperformed established creatinine and/or cystatin C-based equations (CKD-EPI_2009_, CKD-EPI_2012_, CKD-EPI_Cys_, and EKFC; [1,2,3]) in terms of bias and accuracy toward mGFR. The median bias of GFR_NMR_ remained consistently close to zero over the entire eGFR range, whereas the other equations exhibited greater fluctuations and tended to systematically underestimate mGFR especially for eGFR below 60 mL/min/1.73 m^2^. The accuracy of GFR_NMR_ was significantly improved compared to that of other equations, both overall and especially at eGFR < 60 mL/min/1.73 m^2^, regardless of the limits of agreement with mGFR chosen (P15, P20, or P30, respectively). The fact that GFR_NMR_ performed better than other equations at narrower error bounds (P15 or P20) is of high clinical relevance because P30, which has previously been used as a standard measure of accuracy [1,2,3], has recently been questioned as an unacceptably wide margin of error [6]. Improved performance of GFR_NMR_ was also reflected by its ability to better reclassify CKD staging than the other equations did, compared to mGFR-based CKD staging. Therefore, our data suggest that GFR_NMR_ is an accurate and robust GFR equation in the adult population with and without CKD and provides clear benefits for patients at risk for inaccurate serum creatinine-based GFR estimates due to increased age or BMI.

As the risk of death increases with diagnosis of CKD associated with an estimated GFR below 60 mL/min/1.73 m^2^ [41], KDIGO recommends using serum creatinine-based GFR estimation for initial assessment of GFR and to use additional tests (such as cystatin C or clearance method) for confirmatory testing once eGFR is <60 mL/min/1.73 m^2^. GFR estimates making use of cystatin C in addition to serum creatinine are more powerful predictors of clinical outcomes than creatinine-only eGFR [42]. The advantage of equations that comprise cystatin C includes greater prognostic values for mortality and cardiovascular disease events and is most apparent among individuals with GFR of 45–59 mL/min/1.73 m^2^ [42]. Patients with both serum creatinine eGFR and combined creatinine- and cystatin C-based eGFR < 60 mL/min/1.73 m^2^, had markedly elevated risks for death, coronary vascular disease, and end-stage renal disease endpoints compared to individuals with GFR > 60 mL/min/1.73 m^2^ [42]. However, depending on the chosen equation, studies have shown that as many as 30–60% patients are misclassified by CKD stage [6]. The minimal bias of the new GFR_NMR_ equation is especially suited to prevent CKD confirmation from systematic underestimation, misclassifying many patients with serum creatinine eGFR < 60 mL/min/1.73 m^2^ and mGFR > 60 mL/min/1.73 m^2^ as CKD stage G3 rather than G2.

Once a diagnosis of CKD has been established, KDIGO, the National Kidney Foundation and the American Diabetes Association recommend assessing CKD progression based on eGFR classification (“CKD stage”). Changes in eGFR leading to changes in GFR classification defines CKD progression and CKD progression establishes the patient’s care plan. Over 70 different eGFR equations have been developed to this end [6]. However, the eGFR values resulting from these equations often differ from mGFR by ±30% or more. An increasing number of studies are noting populations in which measuring the rate of progression of CKD using eGFR has been found inadequate due to limited eGFR accuracy. For example, in a 2013 study of 449 type 2 diabetic patients with mGFR and eGFR values obtained over a median follow-up period of 4.0 (range 1.8–8.1) years, all estimation formulas failed to provide any reliable estimations of GFR changes over time. Long-term GFR decline was largely and uniformly underestimated in the study group [43]. Here again, an eGFR equation like GRF_NMR_, with very low bias and improved accuracy, might allow a more accurate monitoring of CKD progression and thus a better management of CKD patients.

In terms of clinical utility, GFR_NMR_ performed better than CKD-EPI_2012_ in our study, the latter better than CKD-EPI_2009_, and the latter better than CKD-EPI_Cys_. These results highlight the utility of integrating more than one or two biomarkers. In particular, the emerging metabolite markers myo-inositol and valine, in addition to creatinine and cystatin C helped to improve eGFR calculation and CKD staging in our hands. Thus, our data are consistent with recent reports in the literature indicating the need for multiple filtration markers to improve GFR estimation [4,9] and also support the results of our previous proof-of-concept study [12]. Although the current analysis suggests that the GFR_NMR_ approach is superior to other measures of eGFR, we are not yet able to explain the complex physiological mechanisms of the two new biomarkers, which appear to differ in part from the existing clearance concept. Serum myo-inositol, an essential component of the second messenger inositol phosphate and a uremic toxin, is inversely correlated with GFR [10,44,45,46,47,48,49,50]. On the other hand, valine, the plasma level of which is reduced in CKD as a result of metabolic acidosis, correlates positively with GFR [48,51,52,53,54,55]. These two emerging biomarkers turned out to play an essential role complementing serum creatinine and cystatin C for accurate GFR estimation and outbalanced their deficiencies in unbiased estimation of GFR. In that sense, the new GFR_NMR_ constellation confirmed the results of our proof-of-concept, where myo-inositol, valine, and dimethyl sulfone in combination with serum creatinine accurately reflected GFR in pediatric and adult patients [12]. In contrast to the complexity of the implemented biomarkers’ interplay, the experimental complexity of the NMR-based multi-parametric test is reduced to a minimum by a high degree of analytical automation and intact quality checks [12]. The GFR_NMR_ test reports GFR in conventional units adjusted for body surface area (KDIGO guidelines, https://kdigo.org/guidelines, accessed on 9 July 2021), avoiding the risk of misinterpreting the complex biomarker interplay by health care providers.

The clinical rationale for considering alternative markers of kidney disease progression was also provided by the U.S. National Kidney Foundation and the U.S. Food and Drug Administration in 2012 [56]. Thus, GFR_NMR_ could address a critical need for equations with higher accuracy and precision for all groups of renal patients, depending not only on ethnicity, sex, and age, but especially on etiology, symptoms, CKD stages, histopathologies, and therapies of renal disease, as well as extra-renal comorbidities [9]. Given the large heterogeneity of patients, repeated single measurements of eGFR require a highly accurate eGFR method to develop new paradigms for evaluating the efficacy of therapeutic interventions aimed at assessing CKD progression in subgroups of adult patients. Moreover, more accurate and precise eGFR values are clinically necessary for renal function-specific subsets of the CKD population, for example when using toxic drugs that may have a narrow therapeutic range, or donor evaluation in kidney transplantation [7,8]. We conclude that GFR_NMR_ could also challenge the concept of the costly and time-consuming mGFR measurements in patients with GFR < 60 mL/min/1.73 m^2^.

The described NMR-based ‘biomarker constellation’ approach, combining novel metabolite markers with established filtration markers to more accurately reflect both glomerular filtration rate and CKD-associated renal dysfunction, is a promising extension of the classical concepts of biomarker ‘panels’ or ‘signatures’. The GFR_NMR_ approach should allow the development of eGFR equations that are personalized and tailored to patients and their specific renal diseases, e.g., diabetes mellitus, nephrotic syndrome, tubulopathies, or after renal transplantation. Because these nephropathies directly or indirectly affect the diagnostic quality of markers for GFR estimation, a ‘super constellation’ or ‘etiology-specific GFR biomarker constellations’ might provide the most accurate results. Such an approach will most likely require the quantification of multiple novel renal biomarker components in addition to established markers. As NMR is a multiplex analyzer capable to quantify precisely multiple unlabeled metabolites in a simultaneous physical measurement step, it appears to be especially suited to avoid increasing analytical costs associated with multiple single biomarker assays. Nevertheless, implementing such a ‘super constellation’ will require large cohorts covering all or most relevant CKD etiologies to generate sufficient data and enable powerful machine learning-based modelling and validation–a major hurdle for individual research groups, requesting future collaborative approaches.

Despite the numerous discussed strengths of the new GFR_NMR_ equation, our study has several limitations. First, the adult population used in this study was predominantly Caucasian, and validation of the new GFR_NMR_ equation has yet to be performed in other ethnic groups. Second, the internal validation cohort consisted of samples with heterogeneous exogenous renal clearance markers of mGFR as reference standards compared to the development and external validation cohorts (Table 1). We cannot exclude the possibility that this heterogeneity introduced a bias into our analysis. Third, external validation of GFR_NMR_ should ideally be performed by independent investigators. However, this limitation was compensated using independent internal and external validation datasets during model selection.

## 5. Conclusions

GFR_NMR_ is a patient-friendly diagnostic method with superior accuracy and precision compared to other recommended eGFR equations, both overall and especially at eGFR < 60 mL/min/1.73 m^2^. Furthermore, it can be repeated in short intervals to monitor patients with GFR < 60 mL/min/1.73 m^2^ or patients at risk for inaccurate serum creatinine-based GFR estimates due to increased age or BMI. The GFR_NMR_ equation might assist in a more robust diagnosis and monitoring of CKD progression.

## 6. Patents

ES has a patent application WO002020065092A1 pending.

## Figures and Tables

**Figure 1 diagnostics-11-02291-f001:**
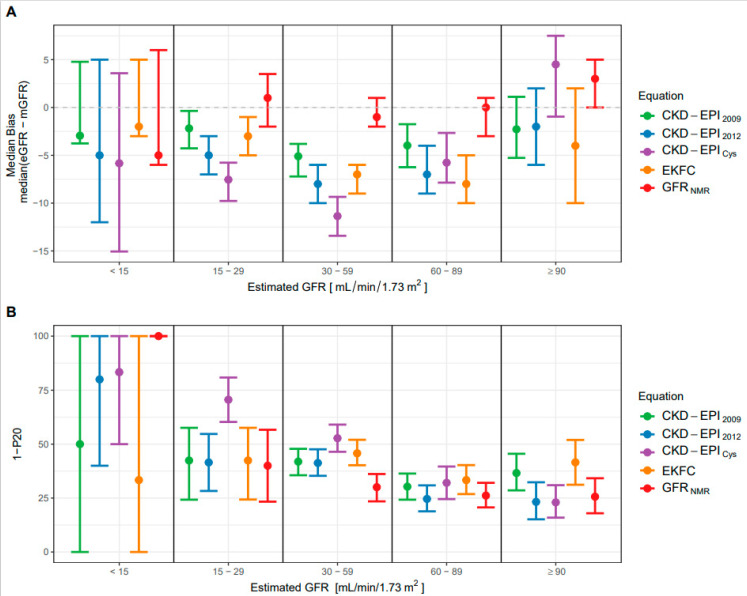
Performance of the eGFR-estimating equations. (**A**) Median bias ± 95% confidence intervals (CI) according to different ranges of eGFR (mL/min/1.73 m^2^ of body-surface area) for the GFR_NMR_, CKD-EPI_2009_, CKD-EPI_2012_, CKD-EPI_Cys_ and EKFC equations. (**B**) Accuracy (1-P20 ± 95% CI) according to different ranges of estimated GFR (mL/min/1.73 m^2^ of body-surface area) for the GFR_NMR_, CKD-EPI_2009_, CKD-EPI_2012_, CKD-EPI_Cys_ and EKFC equations. 1-P20 is the percentage of eGFR values lying outside the tolerance range of 20% of measured GFR. In (**A**,**B**), the range of *n* values per eGFR subgroup are as follows: 3 to 6 for eGFR < 15 mL/min/1.73 m^2^, 30 to 68 for 15–29 mL/min/1.73 m^2^, 213 to 271 for 30–59 mL/min/1.73 m^2^, 159 to 237 for 60–89 mL/min/1.73 m^2^, and 77 to 117 for ≥90 mL/min/1.73 m^2^.

**Figure 2 diagnostics-11-02291-f002:**
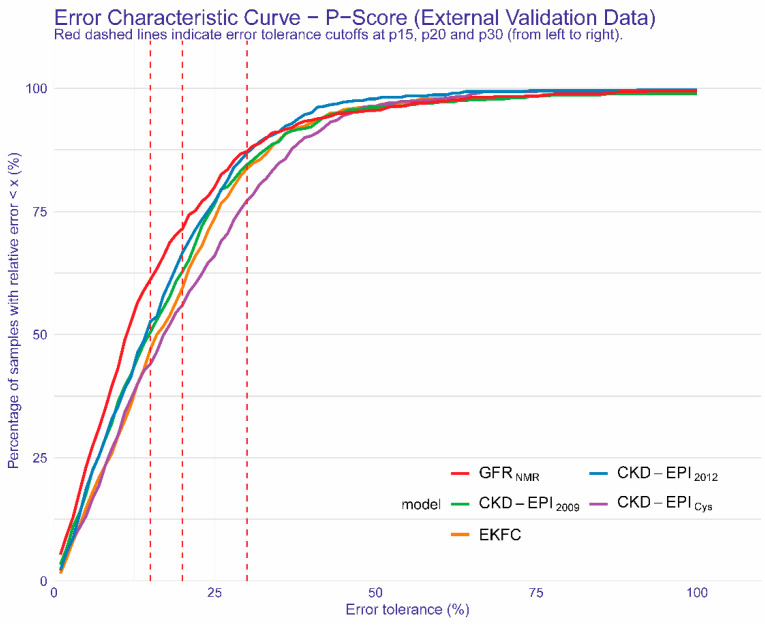
Regression Error Characteristic Curve for all error tolerance Px values for GFR_NMR_, CKD-EPI_2009_, EKFC, CKD-EPI_Cys_, CKD-EPI_2012_. Px denotes the percentage of eGFR values within x% of measured GFR. The red dashed lines indicate error tolerance cutoffs of (from left to right) P15, P20 and P30, respectively. The *y*-axis shows the corresponding percentage of samples within the given error tolerance Px value on the *x*-axis. Colored curves represent the results of the different equations.

**Table 1 diagnostics-11-02291-t001:** Study samples’ characteristics.

	Development Set	Internal Validation Set	External Validation Set	*p*-Value
N	816	439	600	
Age in years, mean ± SD (range)	55 ± 14 (18–86)	58 ± 15 (18–88)	56 ± 14 (19–88)	0.0269 ^1^
Height in cm, mean ± SD (range)	170 ± 10 (142–199)	170 ± 10 (143–199) ^6^	170 ± 10 (121–196)	0.4929 ^1^
Weight in kg, mean ± SD (range)	84 ± 21 (23–195)	80 ± 19 (38–164)	86 ± 21 (31–160)	<0.0001 ^1^
BMI in kg/m^2^, mean ± SD (range)	29 ± 6 (7–68)	28 ± 6 (15–52) ^6^	30 ± 6 (16–58)	0.0005 ^1^
Male, N (%)	431 (52.8%)	143 (55.4%)	334 (55.7%)	0.5479 ^2^
Black, N (%)	17 (2.1%)	5 (1.1%)	17 (2.8%)	0.1575 ^2^
mGFR				
mean ± SD (range) ^5^	67 ± 28 (3–183)	67 ± 28 (5–166)	68 ± 27 (3–166)	0.7280 ^1^
Iothalamate, N (%)	816 (100.0%)	269 (61.3%)	600 (100.0%)	- ^4^
Iohexol, N (%)	0 (0.0%)	92 (21.0%)	0 (0.0%)	
Inulin, N (%)	0 (0.0%)	77 (17.5%)	0 (0.0%)	
^51^Cr-EDTA, N (%)	0 (0.0%)	1 (0.2%)	0 (0.0%)	
CKD stage, N (%)				
G1	170 (20.8%)	92 (21.0%)	135 (22.5%)	0.7989 ^3^
G2	291 (35.7%)	156 (35.5%)	214 (35.7%)	
G3	304 (37.3%)	158 (36.0%)	211 (35.1%)	
G4	46 (5.6%)	26 (5.9%)	36 (6.0%)	
G5	5 (0.6%)	7 (1.6%)	4 (0.7%)	
eGFR				
GFR_NMR_, mean ± SD (range) ^5^	66 ± 25 (7–151)	66 ± 26 (7–175)	67 ± 25 (9–158)	0.5685 ^1^
CKD-EPI_2009_, mean ± SD (range) ^5^	63 ± 26 (5–165)	68 ± 28 (4–146)	64 ± 24 (8–142)	0.0027 ^1^
CKD-EPI_2012_, mean ± SD (range) ^5^	60 ± 26 (6–144)	63 ± 27 (6–145)	62 ± 25 (8–150)	0.1261 ^1^
CKD-EPIc_ys_, mean ± SD (range) ^5^	59 ± 28 (7–136)	60 ± 28 (7–148)	61 ± 28 (9–144)	0.3902 ^1^
EKFC, mean ± SD (range) ^5^	61 ± 23 (6–130)	65 ± 25 (5–136)	62 ± 22 (8–120)	0.0074 ^1^

^1^ Two sided ANOVA; ^2^ Fisher’s exact test; ^3^ Pearson’s Chi-squared test; ^4^ For the comparison of used mGFR tracer no classical test was available, as the tracer method for the Internal Validation Set was different by design (mixture of European and US cohort); ^5^ mL/min/1.73 m^2^ of body-surface area; ^6^ Calculated from 399 samples, due to 40 samples with missing height data. Abbreviations: BMI, body mass index; CKD, chronic kidney disease; CKD-EPI, Chronic Kidney Disease Epidemiology Collaboration equations (CKD-EPI_2009_ [1], CKD-EPI_2012_ and CKD-EPIc_ys_ [2]); eGFR, estimated GFR; EKFC, European Kidney Function Consortium equation [3]; GFR, glomerular filtration rate; mGFR, measured GFR.

**Table 2 diagnostics-11-02291-t002:** GFR_NMR_ equation according to serum cystatin C cutoff levels and sex.

Sex	Serum Cystatin C [mg/L]	GFR_NMR_ Equation ^1^
Female	<1.02	238 × cystatin C^−0.4114^ × creatinine^−0.3798^ × valine^0.1628^ × 0.9979^myo_inositol^ × 0.9963^age^
	≥1.02	239 × cystatin C^−0.6443^ × creatinine^−0.3798^ × valine^0.1628^ × 0.9979^myo_inositol^ × 0.9963^age^
Male	<1.22	266 × cystatin C^−0.5867^ × creatinine^−0.3798^ × valine^0.1628^ × 0.9979^myo_inositol^ × 0.9963^age^
	≥1.22	269 × cystatin C^−0.6419^ × creatinine^−0.3798^ × valine^0.1628^ × 0.9979^myo_inositol^ × 0.9963^age^

^1^ Coefficients were rounded to four digits; the sex- and cutoff-dependent intercept was rounded to 0 digits; coefficients are brought to the scale of the respective metabolite serum concentration (µmol for valine, creatinine and myo-inositol; mg/L for cystatin C).

**Table 3 diagnostics-11-02291-t003:** Performance of eGFR-estimating equations in the overall external validation set (*n* = 600) and according to eGFR ranges.

Variable		Estimated GFR Range		
	Overall	<60	60–89	≥90
	*n* = 600	mL/min/1.73 m^2^ of Body-Surface Area		
Bias—median difference (95% CI) [mL/min/1.73 m^2^]				
CKD-EPI_2009_	−4.0 (−4.8; −2.9) ****	−4.6 (−5.8; −3.3)	−4.0 (−6.2; −1.7)	−2.3 (−5.3; 1.1)
CKD-EPI_2012_	−6.0 (−7.0; −5.0) ****	−7.0 (−8.0; −5.0)	−7.0 (−9.0; −4.0)	**−2.0** (−6.0; 2.0)
CKD-EPI_Cys_	−7.0 (−8.1; −5.8) ****	−10.1 (−11.5; −8.2)	−5.8 (−7.9; −2.7)	4.5 (−1; 7.5)
EKFC	−6.5 (−8.0; −5.0) ****	−6.0 (−8.0; −5.0)	−8.0 (−10.0; −5.0)	−4.0 (−10.0; 2.0)
GFR_NMR_	**0.0** (−1.0; 1.0)	**−1.0** (−2.0; 1.0)	**0.0** (−3.0; 1.0)	3.0 (0.0; 5.0)
Precision—IQR of the difference (95% CI) [mL/min/1.73 m^2^]				
CKD-EPI_2009_	16.4 (15.1; 17.7) *	14.4 (12.5; 15.9)	18.5 (15.5; 22.3)	25.6 (20.4; 34.6)
CKD-EPI_2012_	14.0 (12.0; 16.0)	**11.0** (10.0; 13.0)	16.0 (13.0; 19.0)	21.5 (14.5; 27.5)
CKD-EPI_Cys_	18.1 (16.4; 19.9) *	14.4 (12.4; 16.1)	20.4 (16.3; 22.9)	23.3 (15.5; 27.3)
EKFC	17.0 (15.0; 19.0) *	15.0 (12.0; 16.0)	20.0 (17.0; 24.0)	32.0 (22.0; 42.0)
GFR_NMR_	**13.0** (10.0; 14.0)	**11.0** (10.0; 13.0)	**15.0** (13.0; 19.0)	**20.0** (15.0; 26.0)
Error—mean absolute error (95% CI) [mL/min/1.73 m^2^]				
CKD-EPI_2009_	11.9 (11.0; 12.7) ****	9.4 (8.5; 10.4)	12.8 (11.3; 14.4)	16.6 (14.2; 19.3)
CKD-EPI_2012_	11.1 (10.3; 11.9) ****	9.6 (8.8; 10.6)	12.0 (10.6; 13.5)	**14.0** (11.7; 16.5)
CKD-EPI_Cys_	13.3 (12.3; 14.2) ****	12.9 (11.7; 14.1)	13.2 (11.4; 14.9)	14.4 (12.1; 16.9)
EKFC	12.8 (11.9; 13.7) ****	10.3 (9.3; 11.2)	14.3 (12.6; 16.1)	18.5 (15.5; 21.6)
GFR_NMR_	**10.0** (9.2; 10.8)	**7.6** (6.8; 8.5)	**10.6** (9.4; 11.8)	14.1 (11.9; 16.4)
Accuracy—1-P15 (95% CI) [%]				
CKD-EPI_2009_	49.5 (45.0; 53.7) ****	54.1 (48.6; 60)	44.4 (37.9; 51.0)	46.4 (37.5; 55.4)
CKD-EPI_2012_	47.3 (43.2; 51.5) **	57.7 (52.3; 63.2)	37.7 (30.9; 45.0)	**33.3** (24.2; 43.4)
CKD-EPI_Cys_	56.0 (51.8; 60.0) ****	68.3 (63.1; 73.5)	43.4 (35.2; 51.6)	38.1 (29.2; 46.9)
EKFC	53.0 (49.0; 57.0) ****	57.7 (52.1; 63.2)	46.8 (40.3; 53.7)	51.9 (40.3; 62.3)
GFR_NMR_	**38.8** (34.3; 42.5)	**43.5** (37.4; 49.6)	**35.4** (29.5; 41.4)	35.9 (27.4; 44.4)
Accuracy—1-P20 (95% CI) [%]				
CKD-EPI_2009_	37.2 (33; 41.3) ***	42.1 (36.6; 47.9)	30.3 (24.2; 36.4)	36.6 (28.6; 45.5)
CKD-EPI_2012_	33.3 (29.3; 37.2) *	41.9 (36.1; 47.4)	**24.6** (18.8; 30.9)	**23.2** (15.2; 32.3)
CKD-EPI_Cys_	44.0 (39.7; 48.0) ****	57.0 (51.5; 62.5)	32.1 (24.5; 39.6)	23.0 (15.9; 31.0)
EKFC	40.5 (36.3; 44.7) ****	45.3 (39.7; 51.1)	33.3 (26.9; 40.3)	41.6 (31.2; 51.9)
GFR_NMR_	**28.5** (24.7; 32.2)	**32.1** (26.4; 37.8)	26.2 (20.7; 32.1)	25.6 (17.9; 34.2)
Accuracy—1-P30 (95% CI) [%]				
CKD-EPI_2009_	15.5 (12.5; 18.7)	17.6 (13.4; 22.1)	13.1 (8.6; 18.2)	14.3 (8.0; 21.4)
CKD-EPI_2012_	13.2 (10.2; 16.0)	17.7 (13.5; 22.3)	**7.9** (4.2; 12.0)	**9.1** (4.0; 15.2)
CKD-EPI_Cys_	22.8 (19.3; 26.2) ****	32.6 (27.4; 38.1)	11.9 (6.9; 17.0)	9.7 (4.4; 15.9)
EKFC	16.3 (13.2; 19.5)	18.9 (14.7; 23.4)	13.9 (9.3; 18.5)	13.0 (5.2; 20.8)
GFR_NMR_	**12.8** (9.8; 15.5)	**17.5** (13.0; 22.4)	9.3 (5.5; 13.1)	10.3 (5.1; 16.2)

Bold numbers highlight the best performance results in each analysis. 1-P15, 1-P20 and 1-P30 denote the percentage of eGFR values lying outside the tolerance range of 15%, 20% and 30% of measured GFR, respectively. Symbols *, **, *** and **** indicate the level of significance for *p*-values < 0.05, <0.01, <0.001 and <0.0001, respectively, in the pairwise tests against GFR_NMR_ for each KPI. Abbreviations: CI, confidence interval; CKD-EPI, Chronic Kidney Disease Epidemiology Collaboration equations (CKD-EPI_2009_ [1], CKD-EPI_2012_ and CKD-EPIc_ys_ [2]); EKFC, European Kidney Function Consortium equation [3]; GFR, glomerular filtration rate; IQR, interquartile range.

**Table 4 diagnostics-11-02291-t004:** Net reclassification by GFR_NMR_ vs. CKD-EPI_2009_ or CKD-EPI_2012_ in the overall external validation set (*n* = 600) and according to eGFR ranges and CKD Stages.

			Estimated GFR Range (mL/min/1.73 m^2^)					
		Overall	<15	15–29	30–44	45–59	60–89	≥90
CKD stage		-	G5	G4	G3b	G3a	G2	G1
Observed mGFR range (mL/min/1.73 m^2^)		3–166	3–14	16–29	30–44	45–59	60–89	90–166
Number of samples		600	4	36	79	132	214	135
**CKD-EPI_2009_**	Total reclassified, N (%)	218 (36.4)	0 (0.0)	10 (27.8)	19 (24.1)	64 (48.5)	76 (35.5)	49 (36.3)
	Correctly reclassified, N (%) ^1^	133 (22.2)	0 (0.0)	6 (16.7)	8 (10.1)	38 (28.8)	54 (25.2)	27 (20.0)
	Incorrectly reclassified, N (%) ^2^	85 (14.2)	0 (0.0)	4 (11.1)	11 (13.9)	26 (19.7)	22 (10.3)	22 (16.3)
	NRI (%)	8.0	0.0	5.6	−3.8	9.1	14.9	3.7
**CKD-EPI_2012_**	Total reclassified, N (%)	168 (28.0)	1 (25.0)	8 (22.2)	32 (40.5)	59 (44.7)	49 (22.9)	19 (14.1)
	Correctly reclassified, N (%) ^1^	107 (17.8)	0 (0)	1 (2.8)	16 (20.3)	36 (27.3)	40 (18.7)	14 (10.4)
	Incorrectly reclassified, N (%) ^2^	61 (10.2)	1 (25.0)	7 (19.4)	16 (20.3)	23 (17.4)	9 (4.2)	5 (3.7)
	NRI (%)	7.6	−25.0	−16.6	0.0	9.9	14.5	6.7

^1^ CKD Stage according to GFR_NMR_ is the same as the one according to mGFR, while CKD-EPI-based CKD stage is different; ^2^ CKD Stage according to CKD-EPI is the same as the one according to mGFR, while GFR_NMR_-based CKD stage is different. Abbreviations: CKD, chronic kidney disease; CKD-EPI, Chronic Kidney Disease Epidemiology Collaboration equations (CKD-EPI_2009_ [1], CKD-EPI_2012_ [2]); GFR, glomerular filtration rate; mGFR, measured GFR; NRI, net reclassification index; N, number of samples.

## Data Availability

The data presented in this study are available within the article or Appendix A.

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
