# Peer review of "Estimating Glomerular Filtration Rate from Serum Myo-Inositol, Valine, Creatinine and Cystatin C"

_diagnostics, 2021, doi:10.3390/diagnostics11122291_

Round 1
Reviewer 1 Report
Authors well describe a new method to estimate eGFR. Their method have shown to be better than currently used also in a external validation cohort. I think that authors should be commended for their work. I have only one major criticism:
1. authors did not provide any information about liver function and why patients had impaired renal function. It might be possible that those with impaired liver function might show lower accuracy.
Author Response
The reviewer is pointing out an important clinical limitation of serum creatinine based GFR estimation. Cirrhosis of the liver with or with concomitant ascites belongs to a variety of clinical circumstances where patients with decreased GFR have a worse prognosis than those without (KDIGO guidelines, https://kdigo.org/guidelines). Hence, accurately evaluating renal function in patients with cirrhosis remains a significant unmet clinical need as serum creatinine alone based eGFR appears to systematically underestimate renal dysfunction. In addition, evolution of diuretic-sensitive ascites to diuretic-refractory ascites and hepatorenal syndrome, is a recognized continuum of renal dysfunction in cirrhosis [1].
We performed a preliminary subgroup analysis of the external validation cohort with patients affected end-stage liver disease (ES-) LD (n=86). GFRNMR tended to displayed lowest MAE as well as highest P15 and P20 values compared to all other equations, reaching significance for the comparison with CKD-EPICys equation (adj. p-values <0.01 for all key performance indicators). Altered hemodynamics in ESLD is reported to result in overestimation of GFR when using creatinine based GFR estimating equations [1]. However, in our external validation cohort we observe underestimation for all equations, slightly for CKD-EPI2009, EKFC and GFRNMR and more pronounced for CKD-EPI2012 and CKD-EPICys. In summary, GFRNMR provides more accurate evaluation of renal function in patients with cirrhosis. However, the unexpected systematic underestimation of all equations warrants further analysis, considering more extensive clinical description of the patient subgroup. As this would be beyond the scope of this publication, we decided to report this subgroup analysis in a separate publication and to provide the preliminary results for the reviewer’s decision making only. Hence, there are no changes to the manuscript itself.
Formula |
MAE |
Median Bias |
P15 |
P20 |
CKD-EPI2009 |
10.8 [8.7 - 12.9] |
-0 [-3 - 4] |
54.7 [43.1 - 65.1] |
60.5 [50.0 – 70.0] |
CKD-EPI2012 |
11.8 [9.7 - 13.9] |
-8 [-10 - -4] |
44.2 [33.7 - 54.7] |
58.1 [46.5 - 68.6] |
CKD-EPICys |
15.7 [12.9 - 18.7] |
-12 [-15- -9] |
31.4 [22.1 - 40.7] |
41.9 [31.4 - 52.3] |
EKFC |
10.7 [8.8 - 12.6] |
-3 [-5 - 0] |
52.33 [40.7 - 62.8] |
60.5 [50.0 – 70.0] |
GFRNMR |
10.6 [8.5 - 12.7] |
-2 [-5 - 3] |
57.0 [45.4 - 67.4] |
66.3 [55.8 - 76.7] |
References
- Mindikoglu, A.L.; Pappas, S.C. New Developments in Hepatorenal Syndrome. Clinical gastroenterology and hepatology : the official clinical practice journal of the American Gastroenterological Association 2018, 16, 162-177 e161, doi:10.1016/j.cgh.2017.05.041.
Reviewer 2 Report
Stämmler et al. proposed a new equation for estimating eGFR that includes valine and myoinositol. The new equation was shown to have only a small difference from mGFR compared to some existing estimators.
The submitted manuscript is very well written.
However, it contains several points that need to be corrected.
1. Recently, a new estimating equation has been proposed. Compared to this formula, is the formula proposed in this study better?
https://www.nejm.org/doi/full/10.1056/NEJMoa2102953
2. The description of how to divide the data set is insufficient. Why does only the internal validation set include measures other than Iothalamate?
3. Also, please add ethnicity information to Table 1.
4. The authors' method includes a new substance to be measured in the equation. This disadvantage (cost and complexity) should be explained.
Author Response
- Recently, a new estimating equation has been proposed. Compared to this formula, is the formula proposed in this study better?
https://www.nejm.org/doi/full/10.1056/NEJMoa2102953
We thank the reviewer for highlighting this very recent and highly relevant publication. A preliminary comparison of the newly suggested combined creatinine and cystatin C equation (CKD-EPI2021) can be found here:
Performance measure |
CKD-EPI2021 |
GFRNMR |
correct CKD stage |
359 (59.9%) |
378 (63.1%) |
NRI |
4.4% |
n.a. |
median bias [ml/min/1.73m²] |
-4 [-5 to -3]a |
0 [-1 to 1]a |
P15 [%] |
57.4 [53.9 to 61.3] b |
61.3 [57.6 to 65.1] b |
a: Wilcoxon Rank Test p<0.0001, b: McNemar ChiSquare Test p=0.08
In summary, GFRNMR seems to provide a more accurate evaluation of renal function in the external validation set (n=600) with an increased proportion of correctly classified CKD stages, a positive NRI, a minimized overall median bias and increased P15 value. This warrants further investigation. As these equations we introduced to enable GFR estimation without an ethnicity term, we see a special need for evaluating the possible changes in diagnostic performance of the new CKD-EPI2021 equations not only compared to GFRNMR, but also to CKD-EPI2009 and CKD-EPI2021 in black and non-black patient cohorts. However, as we have currently very limited access to black patient samples (see also comment 3 of this reviewer), we feel that a meaningful analysis would need substantially increased patients numbers. However, this is clearly beyond the scope of this publication. Hence, we provide these preliminary results for the reviewer’s decision making at this stage, but refrain from direct changes to the manuscript.
- The description of how to divide the data set is insufficient. Why does only the internal validation set include measures other than Iothalamate?
The data of the Mayo cohort was split into three data sets (development, internal and external validation sets) stratified by mGFR range, liver disease status, sex and indication. Following this, we made sure that ethnicity was also distributed quite equally throughout the data sets (exception being the internal validation data set). The Lyon and Berlin data set was considered a valuable data set, however, it was using heterogeneous mGFR tracers, rather than just iothalamate as in the Rochester cohort. Due to the heterogeneous nature of this data set, we found it to be especially valuable during optimization of the models and therefore was utilized in the internal validation data set. This decision was backed by the fact, that we observed the CKD-EPI formulas to show larger bias in the Lyon and Berlin data set, compared to the cohort from Rochester. As the external validation set was especially intended to validate the performance of the models independently (without being seen during model selection process) we wanted to minimize the risk to overjudge the superiority of our equation against CKD-EPI formulas due to a systematic bias in the Lyon and Berlin data set.
We revised our method section 2.1 as follows:
[…] Qualified NMR spectra were obtained from 1,855 serum samples in total. Samples underwent partitioning into “Development”, “Internal Validation” and “External Validation” datasets (Table 1) stratified by mGFR range, liver disease status, sex and clinical indication. The development set was used for equation formulation, training and pre-selection. The internal validation set was applied for selection and internal testing of pre-selected candidate equations. The external validation set was used for confirmation of performance on an independent dataset. The development and the external validation sets consisted of samples from Rochester (n=816 and n=600, respectively) with a homogenous reference standard to minimize potential reference bias, whereas the internal validation set was populated with samples from all three centers (Rochester, n=269; Lyon and Berlin, n=170) with heterogeneous reference methods to maximize generalization of the selected equation (Table 1). This partitioning was supported by the observation, that we observed larger bias of the CKD-EPI equation in samples from Lyon and Berlin, compared to the cohort from Rochester. Hence, the applied partitioning appeared particularity suited to minimize the risk to overjudge superiority of models against CKD-EPI benchmarks due to a systematic bias in the external validation set by samples from Lyon and Berlin.
- Also, please add ethnicity information to Table 1.
We now provide the requested information on ethnicity in Table 1 according to the reviewer comments. Initially we did not include this information, as the data does not contain any statistically relevant proportion of samples from black patients as already addressed in our discussion and supplement.
|
Development Set |
Internal Validation Set |
External Validation Set |
p-value |
N |
816 |
439 |
600 |
|
[…] |
|
|
|
|
Black, N (%) |
17 (2.1%) |
5 (1.1%) |
17 (2.8%) |
0.1575 2 |
[…] |
|
|
|
|
- The authors' method includes a new substance to be measured in the equation. This disadvantage (cost and complexity) should be explained.
As the reviewer correctly pointed out, we aimed at a more complex approach that interprets multiple biomarkers reflecting both glomerular filtration rate and CKD-associated renal dysfunction. We already discuss this point in detail in our recent publication Ehrich et al. [2]. As we outline there, such an approach requires the quantification of several renal biomarkers with high precision and accuracy. To avoid increasing analytical costs associated with multiple single biomarker assays, we applied NMR as a multiplex analyzer capable to precisely quantify creatinine, myo-inositol and valine as multiple unlabeled metabolites in a simultaneous physical measurement step.
The complexity of the approach can be interpreted in two different ways, i.e. experimental complexity and the complex interplay of the four metabolites. As an NMR-based multi-parametric test, GFRNMR is associated to some degree of experimental complexity and the risk of misinterpreting the test results by health care providers. However, the total score of the CLIA categorizations for the GFRNMR test is less than 12, indicating moderate experimental complexity of the test system. This is mainly achieved by a high degree of analytical automation and intact quality checks (see [2] for some details). The interpretation of GFRNMR test results is based on comparison to normative values, which are adjusted for body surface area (BSA), because of the physiologic matching of GFR to kidney size, which is in turn related to BSA. The value of 1.73m² reflects the average value of BSA of 25-year old men and women in the US (KDIGO guidelines, https://kdigo.org/guidelines). While it is known that some populations may have different normal values for BSA, we maintain the 1.73m² value for normalization purposes. Hence, the resulting GFRNMR test reports GFR in conventional units as standard of care does and hence is intuitive to health care providers.
On the other hand, the complex interplay of the four metabolites complementing each other in a way of mitigating individual weaknesses and potentiating their contribution to overall clinical value, represents a physiologically more complex and hence involved way of using biomarkers. The concept expands the previous approach suggested by Levey and co-workers to combine metabolites into a panel to more closely correlate with mGFR [3,4], by additionally reflecting CKD-associated renal dysfunction and co-morbidities. For a detailed discussion see Ehrich et al. [2].
Overall we feel that the NMR approach is especially suited to limited overall analytical costs, that high automation of the AXINONTM system minimizes experimental complexity and that the complexity of the metabolite constellation interprets multiple biomarkers reflecting both glomerular filtration rate and CKD-associated renal dysfunction.
We expanded our discussion section as follows:
[…] Because these nephropathies directly or indirectly affect the diagnostic quality of markers for GFR estimation, a ‘super constellation’ or ‘etiology-specific GFR biomarker constellations’ might provide the most accurate results. Such an approach will most likely require the quantification of multiple novel renal biomarker components in addition to established markers. As NMR is a multiplex analyzer capable to precisely quantify multiple unlabeled metabolites in a simultaneous physical measurement step, it appears to be especially suited to avoid increasing analytical costs associated with multiple single biomarker assays. Nevertheless, implementing such a ‘super constellation’ will require large cohorts covering all or most relevant CKD etiologies to generate sufficient data and enable powerful machine learning-based modelling and validation – a major hurdle for individual research groups, requesting future collaborative approaches.
[…] These two emerging biomarkers turned out to play an essential role complementing serum creatinine and cystatin C for accurate GFR estimation and outbalanced their deficiencies in unbiased estimation of GFR. In that sense, the new GFRNMR constellation confirmed the results of our proof-of-concept, where myo-inositol, valine, and dimethyl sulfone in combination with serum creatinine accurately reflected GFR in pediatric and adult patients [12]. In contrast to the complexity of the implemented biomarkers’ interplay, the experimental complexity of the NMR-based multi-parametric test is reduced to a minimum by a high degree of analytical automation and intact quality checks [12]. The GFRNMR test reports GFR in conventional units adjusted for body surface area (KDIGO guidelines, https://kdigo.org/guidelines), avoiding the risk of misinterpreting the complex biomarker interplay by health care providers.
References
- Mindikoglu, A.L.; Pappas, S.C. New Developments in Hepatorenal Syndrome. Clinical gastroenterology and hepatology : the official clinical practice journal of the American Gastroenterological Association 2018, 16, 162-177 e161, doi:10.1016/j.cgh.2017.05.041.
- Ehrich, J.; Dubourg, L.; Hansson, S.; Pape, L.; Steinle, T.; Fruth, J.; Hockner, S.; Schiffer, E. Serum Myo-Inositol, Dimethyl Sulfone, and Valine in Combination with Creatinine Allow Accurate Assessment of Renal Insufficiency-A Proof of Concept. Diagnostics 2021, 11, doi:10.3390/diagnostics11020234.
- Coresh, J.; Inker, L.A.; Sang, Y.; Chen, J.; Shafi, T.; Post, W.S.; Shlipak, M.G.; Ford, L.; Goodman, K.; Perichon, R., et al. Metabolomic profiling to improve glomerular filtration rate estimation: a proof-of-concept study. Nephrology, dialysis, transplantation : official publication of the European Dialysis and Transplant Association - European Renal Association 2019, 34, 825-833, doi:10.1093/ndt/gfy094.
- Freed, T.A.; Coresh, J.; Inker, L.A.; Toal, D.R.; Perichon, R.; Chen, J.; Goodman, K.D.; Zhang, Q.; Conner, J.K.; Hauser, D.M., et al. Validation of a Metabolite Panel for a More Accurate Estimation of Glomerular Filtration Rate Using Quantitative LC-MS/MS. Clinical chemistry 2019, 65, 406-418, doi:10.1373/clinchem.2018.288092.
Round 2
Reviewer 2 Report
I respect the authors for their appropriate and courteous response.
Author Response
Thank you for your comments